Associations between self-reported and objective face recognition abilities are only evident in above- and below-average recognisers

http://orcid.org/0000-0002-7760-318X Estudillo Alejandro J. 1 2 aestudillo@bournemouth.ac.uk
http://orcid.org/0000-0002-0387-3590 Wong Hoo Keat 2
1 Department of Psychology, Bournemouth University , Bournemouth , UK
2 School of Psychology, University of Nottingham—Malaysia Campus , Semenyih, Selangor , Malaysia
Barnhart Anthony
Electronic publication date: 2021 Jan 11
Publication date: 2021
Volume: 9
Electronic Location ID: e10629
Received 2020 Jun 9; Accepted 2020 Dec 1
Copyright: © 2021 Estudillo and Wong
Copyright year: 2021
Copyright holder: Estudillo and Wong
License: This is an open access article distributed under the terms of the Creative Commons Attribution License, which permits unrestricted use, distribution, reproduction and adaptation in any medium and for any purpose provided that it is properly attributed. For attribution, the original author(s), title, publication source (PeerJ) and either DOI or URL of the article must be cited.
License URL: https://creativecommons.org/licenses/by/4.0/

Keywords: Self-reported face recognition abilities, Developmental prosopagnosia, Individual differences in face recognition, Cambridge face memory test, Prosopagnosia index

Funding: The authors received no funding for this work.

==============================
The 20-Item Prosopagnosia Items (PI-20) was recently introduced as a self-report measure of face recognition abilities and as an instrument to help the diagnosis of prosopagnosia. In general, studies using this questionnaire have shown that observers have moderate to strong insights into their face recognition abilities. However, it remains unknown whether these insights are equivalent for the whole range of face recognition abilities. The present study investigates this issue using the Mandarin version of the PI-20 and the Cambridge Face Memory Test Chinese (CFMT-Chinese). Our results showed a moderate negative association between the PI-20 and the CFMT-Chinese. However, this association was driven by people with low and high face recognition ability, but absent in people within the typical range of face recognition performance. The implications of these results for the study of individual differences and the diagnosis of prosopagnosia are discussed.

Introduction

Face recognition is a very important cognitive skill that enables successful social interactions with peers. Interestingly, despite being a remarkably common process, face recognition presents substantial variation among individuals, and this variation has important theoretical and practical consequences (Lander, Bruce & Bindemann, 2018; Wilmer, 2017). On one side of the distribution, we find people with extraordinary abilities to identify faces, known as super-recognizers (Russell, Duchaine & Nakayama, 2009). Super-recognizers present above normal performance in a variety of face identification tasks, including unfamiliar and familiar face recognition (Russell, Duchaine & Nakayama, 2009), and face matching (Robertson et al., 2016). Given their extraordinary abilities to identify faces, employing super-recognizers can be highly valuable in those applied scenarios whereby the identification of faces is of paramount importance, such as surveillance, eyewitness identification, and ID-verification settings (Ramon, Bobak & White, 2019).

On the other side of the distribution, we find people with severe difficulties to recognize faces. These difficulties can arise following brain injury as in the case of acquired prosopagnosia (Rossion, 2018), or as consequence of atypical brain development as in the case of developmental prosopagnosia (Bowles et al., 2009; Dalrymple & Palermo, 2016; Duchaine & Nakayama, 2006). Although acquired prosopagnosia is an extremely rare disorder (Rossion, 2018), it has been estimated that the prevalence of developmental prosopagnosia is around 2–3% in general population (Barton & Corrow, 2016; Bate & Tree, 2017; Bowles et al., 2009; Dalrymple & Palermo, 2016; Kennerknecht, Ho & Wong, 2008). As consequence of their difficulties identifying faces, people with prosopagnosia find social situations particularly stressful and are prone to depression, anxiety and social avoidance disorders (Dalrymple et al., 2014; Yardley et al., 2008).

The Cambridge Face Memory Test (CFMT) was introduced as an objective tool to study individual differences in face identification (Duchaine & Nakayama, 2006; Russell, Duchaine & Nakayama, 2009). This task can be completed in approximately 20 min and requires the identification of faces across different images of the same person, avoiding the limitations of simple pictorial recognition (Bruce, 1982; Estudillo, 2012; Estudillo & Bindemann, 2014; Longmore, Liu & Young, 2008) and the use of non-facial cues (e.g., make up, clothing, hairstyle). Although the CFMT was initially introduced with Caucasian faces, more recent versions have adapted the face stimuli to Chinese and South East Asian populations: the CFMT-Chinese (McKone et al., 2012, 2017). Remarkably, these two versions of the CFMT are psychometrically quite robust as they present internal reliability scores of between 0.85 and 0.90 (Bowles et al., 2009; Estudillo et al., 2020), which is an important requirement for measures of individual differences.

Although few researchers would disagree about the importance of objective measures to evaluate individual differences in face identification, phenomenological or self-reported measures have attracted the interest of researchers in recent years (Bobak, Mileva & Hancock, 2019; Livingston & Shah, 2018; Palermo et al., 2017; Shah et al., 2015a, 2015b). In self-reported measures of face identification, observers are, generally, asked to rate their level of agreement with a set of statements describing different situations involving face recognition abilities. It has been suggested that these self-reported measures can be used as screening or complementary tools to measure individual differences in face identification and, particularly, in the diagnosis of prosopagnosia (Shah et al., 2015a, 2015b). Although several self-reported measures of face identification have been built (Bate & Dudfield, 2019; Bobak, Mileva & Hancock, 2019; Palermo et al., 2017), the 20-item prosopagnosia index (PI-20) is probably the most widely-used (Shah et al., 2015a, 2015b). This questionnaire is comprised of 20 items in a five-point Likert scale, describing different situations involving face identification (e.g., “My face recognition ability is worse than most people”). Higher scores in the PI20 index worse face recognition skills. Scores in the PI-20 are negatively associated with different objective face identification measures, such as the CFMT original (Livingston & Shah, 2018; Shah et al., 2015a; Ventura, Livingston & Shah, 2018) and the CFMT-Chinese (Estudillo, in press; Nakashima et al., 2020) versions, famous faces recognition tests (Shah et al., 2015b; Ventura, Livingston & Shah, 2018), and the Glasgow Face Matching Test (Shah et al., 2015b). Importantly, this negative association is held in those participants who have not received formal feedback about their face recognition abilities (Gray, Bird & Cook, 2017; Livingston & Shah, 2018). Therefore, it seems that the PI-20 is a fast and valid method that can be used as a complementary tool for studying individual differences in face identification.

However, despite these promising findings, the PI-20 and other self-reported measures of face identification are not free of criticisms. For example, it has been reported that the associations between objective and self-reported measures of face identification are only moderate (Bobak, Mileva & Hancock, 2019; Gray, Bird & Cook, 2017; Shah et al., 2015a). This is such that PI-20 scores explain only around 5–15% of the variance in the scores of the CFMT in normal populations (Gray, Bird & Cook, 2017; Livingston & Shah, 2018; Matsuyoshi & Watanabe, 2020; Nakashima et al., 2020). Interestingly, when developmental prosopagnosics are tested, the amount of explained variance increases to 46% (Shah et al., 2015a), suggesting that compared to normal population, people with prosopagnosia might have more accurate insights into their face recognition abilities (Palermo et al., 2017). In addition, it has been shown that super-recognizers also seem to have better insights into their face recognition abilities compared to control participants, especially in target-present face matching trials (Bate & Dudfield, 2019), although this study did not use the PI-20. Thus, one question that arises is whether the moderate association usually found between objective and self-reported measures of face identification is merely driven by people with relatively low and high objective face recognition abilities.

The present study seeks to shed light on this question using the Mandarin version of the PI-20. Similar to other studies, our observers performed both the PI-20 and the CFMT. In addition to exploring individuals’ insights into face recognition abilities on the entire distribution of scores, unlike other studies, we also explored whether these insights depend on observers’ objective face recognition performance level. To achieve this, we divided our sample into four different quartiles according to their scores in the CFMT. This quartile-split approach is a standard approach in metacognition research that was firstly introduced by Dunning et al. (2003). This method has been widely used since then to study metacognition in different cognitive processes, including reasoning (Pennycook et al., 2017), intelligence (Unsworth & Engle, 2005), working memory (Adam & Vogel, 2017) and, more recently, face perception (Zhou & Jenkins, 2020). The aim of this approach is to have four subgroups of participants of approximately the same size, representing different degrees of performance in the task (i.e., Q1: low performance, Q2: low-average performance, Q3: average-high performance, Q4: high performance). We also applied the quartile-split approach to reanalyze the data of a published study that found a robust association between the CFMT and the PI20 in the general population (Gray, Bird & Cook, 2017). If observers have insights into their face recognition abilities, we would find a negative association between the PI20 and the CFMT in the whole sample. If these insights are presented across the whole range of face recognition abilities, this negative association between the PI20 and the CFMT will also be observed in each quartile separately.

Materials and Methods

We confirm that we report how all the measures, manipulations and data exclusions in this study. We also report how we have determined our sample size.

Participants

Our sample size was determined a priori based on other studies (Shah et al., 2015b; Ventura, Livingston & Shah, 2018). A total of 280 Chinese ethnicity students from HELP University and the University of Nottingham Malaysia took part in this study for course credits. Twenty-five participants were excluded due to performance at chance level and/or abnormally fast response times (<500 ms), suggesting lack of engagement with the task. Our final web sample consisted of 255 participants (67 males). Observers’ mean age was of 21 years (SD = 4.2). All participants reported having normal or corrected-to-normal vision. Observers were naïve regarding the aims of the study and were never tested before with either the CFMT or the PI-20. Participants provided written informed consent1 and were debriefed at the end of the study. This study was approved by the university research ethics review committee (AJE271017).

Materials, apparatus and procedure

Participants were tested over the web using the application testable (www.testable.com) to present stimuli and to record observers’ responses. This study involves an objective measure of face recognition (i.e., the CFMT-Chinese; McKone et al., 2012) and a self-reported measure of face recognition (i.e., the PI-20; Shah et al., 2015a). The PI-20 was translated into Mandarin. The order of these tasks was randomized across participants.

The CFMT-Chinese. The paradigm of the CFMT-Chinese (McKone et al., 2012) is identical to the classical CFMT (Duchaine & Nakayama, 2006) but it contains Chinese-ethnic faces as stimuli. This task requires participants to learn and recognize different unfamiliar faces in three different stages: same image, novel images and novel images with noise. Observers are firstly required to study a target identity presented in frontal, mid-profile left, and mid-profile right orientations Each of these orientations is presented individually for 3 s. Observers are then presented with the target identity among two other filler face distractors and are required to identify the target, in each of the three orientations. The three face images are presented until response. This procedure is repeated for five additional target identities. The same image stage contains a total of 18 trials (three face orientation for each of the six identities). Observers then proceed to the novel images stage. In this stage, observers are required to study the same six target identities for 20 s. All the target identities are simultaneously presented in the same display. Observers are then presented with a new instance of the target identity among two filler face distractors and are asked to identify the target face. On each 3-item stimulus array, the target face can be any one of the six learned targets, always presented in a novel image (i.e., different viewpoints, lighting condition or both). This second stage has a total of 30 trials. The novel images with noise stage is identical to the novel images stage, but target identities and filler faces distractors are presented with visual noise to make the task harder. This stage has 24 trials. The maximum total scores observers can get in the CFMT is 72 (i.e., one point for each correct trial). Internal reliability analysis showed an alpha value of 0.85 which is in agreement with previous research (Estudillo et al., 2020; Estudillo, in press; McKone et al., 2012).

The Mandarin PI-20. In this stage, observers completed the Mandarin version of the PI-20 (see Appendix 1). The PI-20 (Shah et al., 2015a) is a self-reported measure of face recognition. It contains 20 items describing daily life situations related with face recognition (e.g., My face recognition ability is worse than most people). Observers are required to rate their agreement with each statement on a five-point Likert-scale (1 = strongly agree, 5 = strongly disagree). Items 8, 9, 13, 17 and 19 were reverse scores. Lower scores in the PI-20 indicates lower face recognition abilities. Internal reliability analysis revealed an alpha value of 0.88, which is in agreement with previous research (Estudillo, in press; Shah et al., 2015a).

Results

We firstly explored observers’ insights into their face recognition abilities. As shown in Fig. 1A, observers scores in the CFMT-Chinese were negatively associated with their scores in the PI-20 (r = −0.35, p < 0.001, CI [−0.46 to −0.24]). This moderate correlation shows that around 12% of the variation in the CFMT scores can be explained by the scores in the PI-20.

Figure 1 (A) Associations between PI20 scores and performance on the CFMT-Chinese. (B) Associations between PI20 scores and performance on the CFMT-Chinese for each quartile.

Secondly, we explored whether the insights into face recognition abilities are stable across different levels of recognition performance. To achieve this aim, observers were grouped in four quartiles, following their scores in the CFMT-Chinese (see Table 1), so two different participants with the same scores in the CFMT will be always allocated to the same quartile. When participants obtained a score that is between the upper and lower limits of two quartiles (e.g., 50), by default, our function will allocate that group of participants to the lower quartile (i.e., the score 50 is allocated to the first quartile, see Table 1)2 . The range of scores were 32–50, for the first quartile; 51–56, for the second quartile; 57–63, for the third quartile; and 64–72, for the fourth quartile. As shown in Fig. 1B, observers’ scores in the CFMT-Chinese were negatively associated with their scores in the PI-20 for the first (r = −0.26, p = 0.03, CI = [−0.47 to −0.02]) and fourth (r = −0.28, p = 0.02, CI [−0.50 to −0.04]) quartiles. Despite these reliable associations, only approximately 7% of the variation in the CFMT scores can be explained by the scores in the PI-20. For the second and third quartiles, the association between the CFMT-Chinese and the PI-20 was not reliable (Q2: r = −0.06, p = 0.57, CI [−0.30 to 0.17], Q3: r = −0.00, p = 0.96, CI [−0.25 to 0.24]). It is possible that the lack of correlation in the second and third quartiles is due to a lack of variation in the data. In fact, a closer inspection of Fig. 1B reveals that this explanation is plausible, especially for the second quartile. To rule out this possibility, we increased the variability of the data by combining scores in these two quartiles. However, the association between CFMT-Chinese and the PI-20 was still not reliable (r = −0.00, p = 0.99, CI [−0.17 to 0.17]). Altogether our results suggest that, at the best, only above- and below-average recognisers have insights into their face recognition abilities.

Table 1 Descriptive statistics for the total sample and across each quartile in our study and Gray, Bird & Cook’s (2017) study.

CFMT Quartile	Present study	Gray, Bird & Cook’s (2017) study	
N	PI-20	CFMT-Chinese	N	PI-20	CFMT	
Mean	SD	Range	Mean	SD	Range	Mean	SD	Range	Mean	SD	Range	
Q1	66	53.81	11.87	31–77	46.07	3.62	32–50	120	47.16	10.46	24–74	43.96	4.08	33–49	
Q2	68	47.10	10.97	29–75	54.04	1.65	51–56	102	40.14	8.94	24–64	54.48	2.48	50–58	
Q3	60	47.60	11.35	28–75	59.51	2.07	57–63	110	38.63	8.38	20–66	61.57	1.71	59–64	
Q4	61	42.65	10.35	27–68	67.40	2.46	64–72	93	37.51	8.49	20–61	67.67	2.01	65–72	
Total	255	47.89	11.78	27–77	56.46	8.19	32–72	425	41.16	9.92	20–74	56.23	9.34	33–72	

Re-analysis of Gray, Bird & Cook’s (2017) study

Gray, Bird & Cook (2017) are freely available (see their Supplemental Data). Their study presented the results of two independent samples (n = 142, and n = 283). We decided to reanalyse Gray, Bird & Cook (2017) results as their procedure is highly similar to ours. As the only remarkable difference between Gray, Bird & Cook (2017) samples is that they were collected in different cities of the UK, we decided to combine them (n = 425 participants, 162 males). This approach has two main advantages. First, it increases the power to detect a potential effect if that effect truly exists. This is particularly important for the quartile-split analysis, as the total sample size is reduced. In addition, as the quartile-split approach takes into consideration the whole range of scores to create the quartiles, the larger the sample size the more certain we are that a specific score corresponds to a specific quartile in the population.

As Gray, Bird & Cook (2017) reported (see Fig. 2A), scores in the CFMT were negatively associated with scores in the PI-20 (r = −0.39, p < 0.001, CI [−0.47 to −0.31]). This moderate correlation is consistent with our results and shows that around 15% of the variation in the CFMT scores can be explained by the scores in the PI-20. Interestingly, when their observers were grouped into quartiles according to their scores in the CFMT (see Fig. 2B; Table 1), there was a negative association between the CFMT and the PI-20, for the first (r = −0.30, p < 0.001, CI [−0.45 to −0.13]) and fourth (r = −0.21, p = 0.03, CI [−0.39 to −0.01]) quartiles. Variation in the CFMT scores explains around 9% and 4% of the scores in the PI-20, for the first and fourth quartile, respectively. Although there was no association between the CFMT and the PI-20 for the second quartile (r = −0.01, p = 0.91, CI [−0.20 to 0.18]), there was a positive reliable association between the CFMT and the PI-20 for the third quartile (r = 0.21, p = 0.02, CI [0.03–0.38]). This association, which is in the opposite direction to the expected if observers had insights into their recognition abilities, disappears when scores in the second and third quartiles are combined (r = −0.00, p = 0.63, CI [−0.16 to 0.10]). Overall, the re-analysis of Gray, Bird & Cook (2017) data is in line with our hypothesis that only below- and above-average recognizers have insights into their face recognition abilities.

Figure 2 Reanalysis of Gray, Bird & Cook (2017) results.

(A) Associations between PI20 scores and performance on the CFMT. (B) Associations between PI20 scores and performance on the CFMT for each quartile.

Discussion

This study investigated observers’ insights into their face recognition abilities with the Mandarin version on the PI-20. We found a reliable negative association between observers’ scores in the CFMT-Chinese and their self-reported face recognition abilities on the PI-20. We also explored whether these insights are consistent across different levels of objective face recognition performance. To achieve this, following previous research in metacognition (Dunning et al., 2003), we adopted a quartile-split approach. We found a weak but reliable negative association between the CFMT-Chinese and the PI-20 in the first and fourth quartiles, but not in the second and third quartiles. We also re-analysed a publicly available sample of 425 Caucasian participants (Gray, Bird & Cook, 2017). In the first and fourth quartile, we found a small but significant negative association between the CFMT and the PI20. In the second quartile, no association was found between both measures. Finally, although in the third quartile we found a positive association between the CFMT and the PI20, this association is in the opposite direction to that expected if participants had insights into their face recognition abilities. Thus, our results not only question previous findings that suggest that adults have moderate to strong insights into their face recognition (Gray, Bird & Cook, 2017; Livingston & Shah, 2018; Shah et al., 2015a), but also suggest that only good and bad recognizers have (limited) insights into their face recognition abilities. It is important to note that the pattern of results found cannot be explained in terms of lack of variation in the scores in the CFMT in the second and third quartiles, as the same pattern of results was observed when the scores in these quartiles were combined. This is remarkable as the range of the CFMT scores in the combined quartiles is similar in size to that in the first quartile and larger than the range of scores in the fourth quartile. This combination of the scores in the second and third quartiles also rules out that our results are due to lack of power, as the number of observations is approximately twice compared to the first and the fourth quartiles.

Some authors have suggested that previously observed associations between objective and self-reported measures of face identification are inflated because those previous studies included developmental prosopagnosic patients in the sample (Bobak, Mileva & Hancock, 2019; Palermo et al., 2017). More recent research showed that this association was held reliable—but much weaker when developmental prosopagnosic patients were not included in the sample (Gray, Bird & Cook, 2017; Livingston & Shah, 2018). Our findings provide compelling evidence suggesting that this association is still mainly driven by people with above- and below-average face recognition abilities.

One question that arises, therefore, is why insights into face recognition abilities are only observed at the lower and upper end of the face recognition abilities distribution. One potential reason could be that these people have previously received formal feedback as part of their participation in face recognition studies (Bobak, Mileva & Hancock, 2019). Yet, in Gray, Bird & Cook (2017) and the current study, observers were naïve regarding the aims of the study and did not complete formal testing of their face recognition ability. In addition, it could also be possible that people with low and high face recognition abilities receive more consistent social feedback about their recognition abilities (e.g., when not recognizing a close friend or when recognizing someone not seen in years). However, this explanation is inconsistent with some reported cases of people with developmental prosopagnosia who were largely unaware of their face recognition deficits (Bowles et al., 2009; Grueter et al., 2007). Thus, why only above- and below-average recognizers have insights into their face recognition abilities is a question for future research.

It must be noted that the aim of the PI-20 is to help the diagnosis of face recognition disorders and particularly prosopagnosia (Gray, Bird & Cook, 2017; Shah et al., 2015a, 2015b). In principle, this is further supported by our results. However, as also shown by our results, variation in the CFMT scores only explained around 7% of the scores in the PI-20, which suggests that even people within the lower range of face identification abilities have very limited insights into their face recognition abilities. In fact, it has been estimated that the PI-20 would fail to detect around 60% of developmental prosopagnosics who would be diagnosed with objective measures of face recognition (Arizpe et al., 2019). For this reason, it is recommended that the diagnosis of prosopagnosia should be mostly based on objective tests and complemented with self-reported measures of face identification (Arizpe et al., 2019; Bobak, Mileva & Hancock, 2019; Palermo et al., 2017).

Conclusions

In summary, the current study reports a moderate negative association between the CFMT and the Mandarin version of the PI-20. This association is in agreement with previous research (Bobak, Mileva & Hancock, 2019; Gray, Bird & Cook, 2017; Livingston & Shah, 2018; Shah et al., 2015b; Ventura, Livingston & Shah, 2018). However, a deeper analysis of our study and the reanalysis of publicly available data (Gray, Bird & Cook, 2017) suggest that this association is mainly driven by people below- and above-average face recognition abilities. Altogether our results suggest that the use of self-reported measures of face identification should be, when possible, complemented with objective measures.

Supplemental Information

Supplemental Information 1 Observers scores and demographic information

Click here for additional data file.

Supplemental Information 2 Supplementary results

Click here for additional data file.

Supplemental Information 3 PI-20 Mandarin version

Click here for additional data file.

Additional Information and Declarations

Competing Interests

Author Contributions

Human Ethics

Data Availability

1 The consent form was provided in English language

2 It is important to note that the same pattern of results was obtained when these participants are allocated to the upper quartile (i.e., the score 50 is allocated to the second quartile, see Supplementary Results)

The authors declare that they have no competing interests.

Alejandro J. Estudillo conceived and designed the experiments, performed the experiments, analyzed the data, prepared figures and/or tables, authored or reviewed drafts of the paper, and approved the final draft.

Hoo Keat Wong conceived and designed the experiments, performed the experiments, authored or reviewed drafts of the paper, and approved the final draft.

The following information was supplied relating to ethical approvals (i.e., approving body and any reference numbers):

The University of Nottingham Malaysia granted ethical approval to carry out this research (AJE271017).

The following information was supplied regarding data availability:

The raw data are available in the Supplemental Files.

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
