# Peer review of "Associations between self-reported and objective face recognition abilities are only evident in above- and below-average recognisers"

_PeerJ, doi:10.7717/peerj.10629_

## Round 0.1 · original submission · Major Revisions

I have received reviews from two experts in the field (Reviewer 1 is Laura Heisick). While Prof. Heisick is largely positive about your manuscript, Reviewer 2 has voiced a set of concerns that I share about the analytical techniques you have employed. Most importantly, your choice to explore correlations within quartiles does not seem to be theoretically grounded and the reanalyses of data from Gray et al. do not support your narrative.

Unless you adopt a fresh, theoretically-motivated strategy for analyzing your data (see suggestions from Reviewer 2), I am not optimistic that a revision will meet threshold for publication at PeerJ, but I do want to offer you the opportunity to address these critiques, if you think it possible.

I also share concerns with Prof. Heisick that your methodological descriptions lack clarity. Prof. Heisick has appended an annotated draft of your manuscript where she highlights some details that need further clarification. There are also typos and grammatical errors throughout the manuscript that should be corrected.

I request that you add a statement to the paper confirming whether, for all experiments, you have reported all measures, conditions, data exclusions, and how you determined your sample sizes. You should, of course, add any additional text to ensure the statement is accurate. This is the standard reviewer disclosure request endorsed by the Center for Open Science [see http://osf.io/project/hadz3]. I include it in every review.

·

Basic reporting

The manuscript is written in clear, professional language and provides sufficient context for the study. The research question is clearly stated. There are minor typos throughout the manuscript (e.g., Line 119, second sentence should begin with Observers’), but these do not detract from readability. I have highlighted these typos in the attached copy of the manuscript. The figures are also very clear and are referenced appropriately in-text. I included this comment in the manuscript, but I think a reference figure for the CFMT-Chinese would be helpful to the reader, as the procedure described is at times confusing.

All raw data can be accessed. The only piece of the data file that is not immediately clear is the column labeled ‘Three Groups.’ This information is not referenced in-text, nor is it identified in the data file. Does this refer to the PI-20 score?

Identifying information is also included. I highlighted these in the document (e.g., Lines 116 – 117, Lines 123 – 124).

Experimental design

The research question addressed in the article is well-defined and easy to identify. The introduction nicely sets the stage for the method described and contains sufficient information to convey how this research contributes to knowledge about the association between self-report and objective measures of face recognition abilities. The Method of the study includes several areas of confusing phrasing or points that would benefit from clarification, noted in the attached document. These areas begin at approximately Line 136 (the CFMT-Chinese description). For example, in Line 138, it is not clear if the target orientations appear 6 times each over the course of the 18 trials, and if the distractor identities also appear in the same orientation or can vary.

Validity of the findings

The data appear to be robust and statistically sound, and the results are laid out clearly. In the attached manuscript, I highlighted Line 160 in the results with the following suggestion. It would be helpful to the reader to include some summary statistics (e.g., average scores for the CFMT and PI-20; how many participants fell into each quartile; how quartiles were divided, as they are not equal intervals) before describing the associations. I also think it would be relevant to highlight low and high scores identified by previous work (e.g., what qualifies as prosopagnosia, according to these tests; what qualifies as a super-recognizer; is there an “average” or “normal” score across either of these two tests?). These comparisons would help when assessing the current study. I highlight these points again in the conclusions because a major focus of the introduction is on prosopagnosia. I also wonder how the data in the current study compare to the re-analyzed data from Gray et al., in broad strokes (means and ranges, for example).

I find the speculation about the explanation for why only above- and below-average recognizers seem to have insight into their abilities very interesting. I think the manuscript would benefit from additional speculation about why these patterns emerge, and the implications it might have for real-world face identification.

As a final note: I wonder how the results of the PI-20 and the CFMT compare with other measures of face recognition ability – for example, those using famous faces, or the Glasgow Face Matching Test. I don’t know how prosopagnosia is typically diagnosed (aside from the noted aim of the PI-20 to help diagnosis of face recognition disorders), but I would be interested to see if other subjective or self-report measures of face recognition show similar associations with objective abilities. To that end, I think it would be worth including if there are any other phenomenological measures of face identification beyond the PI-20, if only as an example for the reader.

Additional comments

Summary
This manuscript represents a study that investigated the association between self-report measures of face recognition abilities, the 20-Item Prosopagnosia Items (PI-20) and objective measures of face recognition abilities, the Cambridge Face Memory Test Chinese (CFMT-Chinese). To examine whether previously observed associations between these tests are driven by low- and high-scoring participants, 280 students completed both tests. Using correlational analyses, the authors show negative associations between PI-20 and CFMT-Chinese performance primarily for individuals who score lower or higher than average.

I have attached an annotated copy of the manuscript. Some of the comments in the PDF will be redundant with what is listed above, but I highlighted areas in which I had questions, comments, or suggestions about the content.

Reviewer 2 ·

Basic reporting

The English is not clear throughout, and there are some problems with references not being included in the reference section. I've listed some examples:

Line 25: ‘typical average’ – not clear; only one of these words is needed.
Line 48: These citations could be more clearly linked to the two statements. Which provide evidence that acquired prosopagnosia is low prevalence, and which provide evidence that developmental prosopagnosia is up to 3%?.
Line 55. ‘This task can be completed in approximately…’
Line 74: It’s unclear what you mean by the sentence “one clear advantage..”. It suggests that the advantage of using self-report is because it doesn’t need adapting for different populations, but that’s not the case, as it needs to be translated.
Line 77: Consider changing ‘famous’ to ‘widely used’ – more scientific.
Line 103: ‘objective’
Line 156: no references are given for Estudillo, 2020, or Estudillo et al., 2020.
Line 153: ‘five-point’
Line 154: ‘reverse scored’
Line 160: ‘we first explored observers’ insights’
Line 161: It’s unnecessary to have the R2 here and throughout when you’ve already said how much of the variation is explained, the R2 is superfluous/repetitive.
Line 174: ‘closer inspection of Figure 1B’
Line 201: ‘on the PI20’
Line 226: ‘someone not seen in years’
Line 244: ‘association’
Line 251: ‘objective’
Figure 2 – it should be the CFMT, not CFMT-Chinese.

Experimental design

The research question is well defined, relevant and meaningful. There are some details missing from the method:

Line 117: How fast was ‘abnormally fast’?
Line 130: How was the pI20 translated into Mandarin? Were there any checks on the validity of the translation?
Line 148: the reliability of the PI20 has been previously given as 0.96, which is considerably higher than the current sample.


Gray et al. actually performed two studies; the data used in the current paper collapses across these two samples. Can you give a justification for doing this?

Validity of the findings

I have some major problems with the approach taken in the analysis. Specifically, the selection of quartiles is arbritary. When looking at the scatter plots, it does not look as though there is a non-linear relationship between the variables. I understand that investigating sections is theoretically driven (in that both good and poor performers could have better insight), but using quartiles doesn’t match up to our definitions of good and poor performers. If one took a random slices of any correlation, there may well be differences between the strength of the relationship. Have the authors thought of fitting non-linear models to the data? Or using a more theoretically driven approach to the partitioning of the data?

Given this, it is particularly problematic that the authors do find a significant correlation in the third quartile of Gray et al.’s paper. The existence of a correlation in this quartile (where participants are neither good nor bad at face processing) directly refutes the conclusions of the paper. Therefore, the conclusions are not supported.

There also seems to be some misunderstanding on how correlations work – reducing the range will always limit the strength of a correlation. In the discussion, the authors say that “only good and bad recognisers have (limited) insights”, and discuss this in the final paragraph of the discussion, suggesting they are over-interpreting the drop in R2 which is driven instead by the reduction of score range.

---

## Round 0.2 · Minor Revisions

Thanks for your thoughtful revision. Reviewer 2 from the first round of review re-examined your paper. While they were generally positive about the changes you made, they had a set of suggestions for improving the piece. If you can address these items (which I think you can), I do not see any barrier to publication.

Reviewer 2 ·

Basic reporting

The manuscript in much improved, I found few grammatical mistakes in this version:
Line 75: "have"
Line 200: "observers' insights"
Line 201: "observers' scores"
Line 255: "of the PI-20"

Experimental design

The added information makes the paper much clearer.

Validity of the findings

I continue to have reservations about the calculation of the correlations/quartiles, which I will attempt to explain.

There are uneven numbers of participants in the groups allocated in both experiments’ quartile split, this is because in the middle of the distribution many of participants share the same score. Can the authors describe and justify their approach to allocating scores into the quartiles? In the current experiment, if participants are randomly split evenly between the nearby groups (such that each group is made of 63/64 participants), the correlations change considerably, suggesting the current findings are not robust to the cut-offs used.

The variability in both scales are important when assessing a correlation - the PI20 also does not have an even range across the quartiles. Can the authors also provide the range of PI20 scores for each quartile in Table 1?

One way to help the authors and readers interpret the correlations would be to include 95% confidence intervals on them.

Additional comments

I appreciate the authors’ rebuttal, and their thoughtful response. I apologise for my misunderstanding of the direction of the effect in Q3 in Gray and colleague’s data.

In the title, the authors are suggesting that the CFMT represents ‘actual’ face recognition ability. However, the CFMT is a match-to-sample test with a reasonably high probability of guessing the correct answers by chance (33% on most trials). The CFMT is an objective face test, but is not a wholly accurate test of face recognition ability ‘in the wild’. The authors should change their framing of this in the title and throughout the manuscript. The fact that the PI20 accounts for a relatively small proportion of the variance of CFMT scores could be due to noise within the CFMT rather than issues with the PI20.

---

## Round 0.3 · accepted · Accept

I am satisfied that all of the reviewers' concerns have been addressed and believe your manuscript will make a valuable contribution to the literature. I am pleased that it will appear in PeerJ. Congratulations.